# Evaluation of Selected Bacteria and Yeast for Probiotic Potential in Poultry Production

**DOI:** 10.3390/microorganisms10040676

**Published:** 2022-03-22

**Authors:** Beverly Dixon, Agnes Kilonzo-Nthenge, Maureen Nzomo, Sarayu Bhogoju, Samuel Nahashon

**Affiliations:** 1Department of Agricultural and Environmental Sciences, Tennessee State University, Nashville, TN 37209, USA; beverlyreadixon@gmail.com (B.D.); mo3nana@gmail.com (M.N.); 2Department of Human Sciences, Tennessee State University, Nashville, TN 37209, USA; akilonzontheng@tnstate.edu; 3School of Medicine, University of Kentucky, Lexington, KY 40506, USA; sarayubhogoju@gmail.com

**Keywords:** poultry, probiotic properties, pH, bile tolerance, intestinal attachment

## Abstract

Performance and efficiency of feed utilization in poultry is highly influenced by gut health, which is dependent on intestinal microbial balance. Probiotics are live microbial feed supplements or viable microorganisms that beneficially affect the host animal by improving its gastrointestinal tract (GIT) microbial balance. However, their mode of action and suitable GIT environment favoring their colonization of the GIT is obscure. The probiotic properties of *Lactobacillus plantarum*, *Bifidobacterium longum*, and *Saccharomyces boulardii* were evaluated. These microbes were tested in vitro against gastrointestinal conditions for survivability and their ability to attach to the intestinal mucosa. The ability of the microbes to tolerate and survive varying pH levels and bile concentrations was assessed. The microbes were challenged with a pH of 2 to 7 for 5 h and bile concentrations of 1 to 3% for 6 hrs. The microbes were sampled hourly to evaluate growth or decline in colony-forming units (CFU). *B. longum*, *L. Plantarum*, and *S. boulardii* exhibited significantly higher CFU (*p* < 0.05) at a pH range of 5 to 7, 4 to 7, and 2 to 7, respectively, when compared with other pH levels. *L. plantarum* had much higher colony-forming units per mL within each pH level, except at pH 2 where *S. boulardii* was the only microbe to survive over time. While *L. plantarum* and *S. boulardii* were able to tolerate the various bile concentrations, *B. longum* and *L. plantarum* showed remarkable ability to attach to the intestinal mucosa and to inhibit pathogenic microbes.

## 1. Introduction

The poultry industry, among other livestock markets, is one of the most consistent economic industries in production and consumption. Considering that the utilization and subsequent consumption of different animal species vary as cultural preferences and religious beliefs are observed [1], poultry is not affected by major religious constraints as is beef or pork. In so being, poultry is generally consumed by all cultural groups of people around the world; hence, the demand for poultry is higher than other animal protein sources. Consequently, this has led to increased production capacity in large-scale, conventional poultry production and associated economic loss due to stressful conditions, diseases, and the deterioration in environmental conditions. As a result, conventional poultry producers employ the use of growth promotants and antibiotics to minimize the potential for bacterial infections and loss [2].

In addition, the increased demand for non-conventional poultry such as free-range and organic poultry has increased the management challenges of poultry production [2,3]. The primary concern, as a result, is increased health issues and the potential exposure of the consumer to foodborne diseases, and especially those that may have acquired antibiotic resistance [4]. Profound questions have therefore arisen concerning the administration of antimicrobial agents as prevention and control for diseases in poultry production. Therefore, the likelihood of discontinuing the use of antibiotics as growth stimulants for poultry, and the apprehension of the side-effects of their use as therapeutic agents, have produced a climate in which the consumers, producers, and feed manufacturers are looking for alternatives. Probiotics are being considered as such an alternative [5,6].

Probiotics are live microbial feed supplements or viable, defined microorganisms in sufficient numbers, which beneficially affect the host animal by improving its intestinal microbial balance [7,8]. The gastrointestinal tract (GIT) is an enormous surface inhabited by a complex and diverse community of microorganisms known as the intestinal microflora (IM). The IM is composed of both pathogenic and non-pathogenic microbes, which when present in an unbalanced ratio, can affect the host negatively or positively [9]. Pathogenic microbes can take advantage of certain gastrointestinal targets, colonizing the GIT and causing various intestinal disorders, while non-pathogenic microbes do the opposite and thus create a balance. Consequently, the IM significantly affects the health of the host by manipulating digestion and nutrient absorption, intestinal morphology, and defense of the host against infection [10,11]. Concurrently, colonizing bacteria that interact with the gastrointestinal mucosa can communicate with the underlying epithelial and mucosal lymphoid elements, an interaction that stimulates host defenses in the gut [12]. Probiotics, the beneficial microbes, play an important and significant role in maintaining a balance among IM.

Careful selection of a probiotic for a specific function in the GIT is important, and when placed in foods, probiotics must remain vigorous in the food until consumption [13]. Different probiotics can function uniquely in the development of intestinal host defenses as an adjuvant of immune responses or to strengthen the mucosal barrier [14]. Previous research has also shown that probiotics can play an active role in the histomorphology of the gastrointestinal tract [15,16,17]. In addition, GIT histomorphology plays a vital role in the health of an organism being that it is the primary location for the transfer of nutrients from digesta. Changes in the properties of the GIT could, therefore, affect the absorption of both dietary and endogenous macromolecules and ions [18].

Amat et al. (1996) [18] suggested that increased villus height is directly related to the increased digestive and absorptive function of the intestine due to increased absorptive surface area, expression of brush border enzymes, and nutrient transport systems. Other investigators found that adding a probiotic containing the *Enterococcus faecium* microorganism to broiler diets increased the jejunal villus height [16] and ileal villus height [19]. Additionally, increased intestinal villi height was reported after the addition of *Bacillus subtilis* in association with prebiotics [20]. Awad et al. (2009) [14] discovered that the supplementation of broilers with either symbiotic (a combination of probiotic and prebiotic) or probiotic increased the villus height and villus height–crypt depth ratio in the duodenum and ileum significantly (*p* < 0.05), suggesting an increased epithelial cell turnover due to the feeding of direct-fed microbes.

Given these enormous beneficial attributes of probiotics in animal nutrition, there are numerous perceived desirable traits for the selection of functional probiotics. According to Todd and Kullen (1999) [21], they must: (1) be a normal inhabitant of the gut; (2) be able to adhere to the intestinal epithelium; (3) overcome potential hurdles, such as the low pH of the stomach, the presence of bile acids in the intestines, and the competition against other microorganisms in the gastrointestinal tract. In developing a probiotics formula, it is of utmost importance that each microbial strain is able to survive and grow at the site where it is supposed to be active. That is, the strain should be able to proliferate and colonize at this specific location, which in this project is the gastrointestinal tract (GIT) of chickens. Therefore, to be effective the microbes must be able to tolerate the low pH and high concentration of both conjugated and deconjugated bile acids, which are present in the GIT of the birds.

Therefore, the objective of this research was to ascertain whether three potential probiotics in animal production *Lactobacillus plantarum*, *Bifidobacteria longum*, and *Saccharomyces boulardii* meet these characteristics. It is essential to identify the conditions that will permit these microorganisms to thrive and enhance protection against the establishment of bacterial pathogens in poultry GIT.

## 2. Materials and Methods

This study was approved by the Institutional Animal Care and Use Committee (IACUC) at Tennessee State University (TSU). The ability of the potential probiotic microbials *Lactobacillus plantarum* and *Bifidobacterium longum* and fungi *Saccharomyces boulardii* to tolerate and survive varying pH and bile concentrations in vitro were evaluated as follows:

### 2.1. Microbes and Culture Conditions

The microbes *Lactobacillus plantarum* (ATCC 8014)*, Bifidobacterium longum* (ATCC 15708), and the fungi *Saccharomyces boulardii* (ATCC 9763) were used in this study. *L. plantarum* and *B. longum* were obtained from American Type Culture Collection (ATCC, Manassas, VA, USA) and *S. boulardii* from Microbiologics Inc. (St. Cloud, MN, USA). *L. plantarum* were inoculated in de Man, Rogosa and Sharpe (MRS) media (Oxoid, Basingstoke, Hampshire, UK) for 12–18 h at 37 °C in an anaerobic chamber with anaerobic conditions (carbon dioxide level between 9% and 13%) created by AnaeroGen^TM^ 2.5 L (Oxoid, Lenexa, KS, USA); *B. longum* in Tryptone, Peptone, Yeast Extract (TPY) medium (containing per liter: tryptone: 10 g; peptone: 5 g; yeast extract: 2.5 g; glucose: 5 g; Tween 80: 1 g; cysteine hydrochloride: 0.5 g; K_2_HPO_4_: 2 g; MgCl·6H_2_O: 5 g; ZnSo_4_·7H_2_O: 0.25 g; CaCl: 0.15 g, and trace of FeCl_3_; pH 7 as in Perez et al. (1998) [22] in a Gloveless Anaerobic Chamber™ (Anaerobe Systems, Morgan Hill, CA, USA) for 18 h at 37 °C, and *S. boulardii* in Sabouraud Dextrose (SB) (Becton, Dickinson and Company, Sparks, MD, USA) media in an aerobic environment at 37 °C. All microbes were stored in a final glycerol concentration of 15% at −80 °C.

### 2.2. In Vitro Tests for Survival of Select Microbes in the Gastrointestinal Tract Environment

#### 2.2.1. Acid Tolerance

With slight modification, the method of Jin et al. (1998) [23] was employed to study acid tolerance. Briefly, *L. plantarum*, *B. longum,* and *S. boulardii* were cultured as previously described. The cultures were then centrifuged at 6000× *g* at 4 °C for 10 min, the pellets washed twice in sterile phosphate buffer saline (PBS, pH 7.2; Sigma, St. Louis, MO, USA), and resuspended in 1 mL of PBS. For each microbe, 100 µL of culture was added separately to 2 mL of sterile PBS suspension in a series of 6 tubes The PBS for each test tube differs in pH values (0.5, 1, 2, 3, 4, 5, 6, and 7). Hydrochloric acid (HCl; 2 M) was used to adjust the pH of the PBS. Tubes were incubated at 37 °C for 0, 1, 2, 3, 4, and 5 h anaerobically (*L. plantarum* and *B. longum*) or aerobically (*S. boulardii*). After the incubation period, the cultures were serially diluted and 100 µL was taken from each tube and cultured on the appropriate agar plates (MRS, TPY, or SB) using the spread plate technique. After the incubation period, viable bacterial colonies were counted. Each assay was replicated twice with triple samples per replication.

#### 2.2.2. Bile Tolerance

The method was adapted from that described by Jin et al. (1998) [22] and Vernazza et al. (2006) [24]. Briefly, *L. plantarum*, *B. longum*, and *S. boulardii* were revived as outlined previously. *L. plantarum* and *B. longum* were inoculated into pre-reduced anaerobically sealed tubes containing 10 mL of MRS and TPY, respectively. *S. boulardii* was inoculated in SB broth. The tubes were incubated at 37 °C. About 100 µL was then inoculated into fresh tubes of 10 mL of the appropriate media broth containing 0, 1, 1.5, 2, and 3% bile or oxgall™ (Sigma). A sample was taken every hour for 6 h and subjected to serial dilutions; then, 100 µL aliquots spread onto agar plates to calculate the CFU/mL. After the incubation period, viable bacterial colonies were counted. There were two replications per assay with triplicate samples per replication.

### 2.3. Assessment of the Ability of Probiotic Microbials to Adhere to Intestinal Epithelial Tissue Microbial Culture Preparation

Microbes were cultured as previously described and the cells were harvested by centrifugation at 6000× *g* at 4 °C for 10 min and washed 3 times in 1 mL of sterile PBS (pH 7.2). Cells were then resuspended in Dulbecco’s Modified Eagle’s Medium (DMEM) to a count of 10^7^ CFU/mL and used immediately.

### 2.4. In Vitro Adhesion Assay

Ten 11-day-old male Ross 308 broiler chicks were given free access to feed and water with a daily treatment of Tetracycline at 10 mg/lbs of bird weight for 5 days by gavage. On day five, three birds were randomly selected and were sacrificed by cervical dislocation. Immediately following, a scalpel was used to open the abdominal cavity and aseptically remove the duodenal loop from the small intestine. These samples were then immersed in DMEM and kept on ice for later use. Then, the tissue segments were washed with 100 mL of sterile PBS (pH 7.2) to remove the contents of the lumen using a 400 mL wash bottle. Then, one end of the duodenal segments was sealed with sterile 35 mm dialysis clamps and inoculated with approximately 1 mL of *B. longum*, *L. plantarum,* or *S. boulardii* culture suspended in DMEM at about 10^7^ CFU/mL or just DMEM for control. The open end of the organ culture was then sealed and the exterior rinsed with PBS. After this, they were immersed in 150 mL of DMEM in a 300 mL flask and incubated for 1 hr at 37 °C with moderate agitation at 270 rpm in a 10% CO_2_ atmosphere.

After incubation, the samples were removed from the flask and flushed with 20 mL PBS, cut longitudinally, and fixed in 2.5% glutaraldehyde in 0.1 M sodium cacodylate buffer, pH 4, at room temperature for 1 hr, then transferred to 4 °C overnight for later evaluation by scanning electron microscopy (SEM).

### 2.5. Sample Preparation for SEM Evaluation

Preparation of sample for SEM evaluation was performed at the Vanderbilt University Medical Center (VUMC, Nashville, TN, USA) Cell Imaging Shared Resource Laboratory (CISRL). Samples were rinsed three times in 0.1 M sodium cacodylate buffer for 5 min each time. Then, a secondary fixation was conducted with a 1% osmium tetraoxide in 0.1 M sodium cacodylate for 1 hr. Again, the samples were washed three times in 0.1 M sodium cacodylate buffer for 5 min each time. The samples were then dehydrated with 30%, 50%, 75%, 85%, 95% ethanol, and 3 times in pure ethanol for 15 min at each ethanol percentage. Following dehydration, the samples were placed into a critical point drier where ethanol was replaced with liquid carbon dioxide, heated to the critical point, and the pressure was bled off slowly. Next, the samples were mounted on stubs and stored in a desiccator overnight. Lastly, the samples were sputter-coated, then evaluated by Quanta^TM^ FEG 250 scanning electron microscope.

### 2.6. Agar Spot Test of Probiotic Microorganisms to Competitively Exclude Pathogenic Bacteria in the Chicken Gastrointestinal Tract

This protocol was adapted from Santini et al. (2010) [25]. Agar plates of MRS, TPY, and SB were used for *L. plantarum*, *B. longum*, and *S. boulardii*, respectively (Table 1). In total, 10 μL of each microbial culture (A 600 of about 0.1 mL) was spotted on their respective plates, and was incubated for 12 to 24 hrs at 37 °C. After strain growth, the plates were overlaid with 10 mL of Nutrient Broth (Oxoid) including 0.7% agar inoculated with a cocktail of pathogenic bacteria comprising *S. pullorun*, *S. typhimurum*, *S. entritidis*, *C. coli*, *C. lari*, *C. jejuni*, *E. faecalis*, *E. faecium,* and *E. coli* O157:H7, *E. coli* ATCC 11775, or *E. coli* ATCC 25922 at 10^7^ CFU/mL. Plates were incubated in conditions required for each pathogen as outlined in Table 1. Formation of clear zones of growth inhibition around *L. plantarum*, *B. longum,* or *S. boulardii* colonies and their diameters were observed and recorded. Inhibition was scored as positive if a diameter of clear zone around the colony was 5 mm or larger [26]. This assay was replicated twice.

### 2.7. Statistical Analysis

All data were evaluated by one-way ANOVA using the generalized linear model (GLM) procedure of SAS (SAS Institute Inc., Cary, NC, USA, 2002) [27]. Means were separated using Duncan’s multiple range test at a 5% level of significance.

## 3. Results and Discussion

The ability for the potential probiotics *Bifidobacterium longum*, *Lactobacillus plantarum*, and *Saccharomyces boulardii* to survive various pH levels is presented in Figure 1. After subjecting these microorganisms to pH levels of 2, 3, 4, 5, 6, and 7 from time 0 to 5 h, *B. longum* displayed higher survivability in pH 4 to pH 7 from 0–5 h. While the *B. longum* colony count was steady above 3 CFU/mL at pH 4–7 and from 0–5 h, there was a slight reduction in the *B. longum* to 2.98 log CFU/mL at pH 4 at 4–5 hr. At pH 3 the colony count was very brief and declined to zero within the first h of incubation, and at pH 2, the *B. longum* did not survive at all. The highest *B. longum* colony count was 3.3 CFU/mL at time 0 h and pH 7, and at pH 4–7, the colony count was 2.98–3.3 log CFU/mL.

Overall, a slight but not statistically significant decline in the bacterial count was observed at all pH levels with an increase in the incubation period, an indication that a continuous supply of *B. longum* in feed would be essential to maintain substantial bacterial count in the gastrointestinal tract of the host. These findings correspond with the report of Biavati et al. (2000) [28] that pointed at the optimum pH for Bifidobacterium being between 6.5 and 7.0. They further noted that no growth was recorded at a pH lower than 4.5 (only *B. thermacidophilum* had a delayed growth at pH 4). However, although decreasing over time, this study found that *B. longum* can survive at pH 4 for at least 5 hrs.

The *Lactobacillus plantarum*, on the other hand, displayed tremendously significant survivability (*p* < 0.05) at pH 4–7 when compared to other microorganisms. Similar to *B. longum*, *L. plantarum* did not do well at pH 2 and pH 3 where the initial bacterial count at time 0 h was 2.48 log CFU/mL and 7.41 log CFU/mL, respectively. At pH 3, however, the *L. plantarum* count decreased over time from 7.41 log CFU/mL to 1.31 log CFU/mL at 5 h post-incubation. Thus, unlike *B. longum*, *L. plantarum* showed more resilience in pH 3 as it survived at that pH level for 5 h over a drastic reduction in colony count after 1 hr. At pH 2, *L. plantarum* did not survive past 0 hr. However, *L. plantarum’s* colony count remained steady at 7.04–8.06 log CFU/mL regardless of exposure time to the various pH levels. It was not a surprise that *L. plantarum* was able to withstand pH levels as low as 3 being that *Lactobacilli* have the unique ability to decompose complex carbohydrate sources into simpler forms and synthesize mainly lactic acid. This characteristic has warranted their use as natural acid-producer bioreactors to replace the production of lactic acid from gasoline and other carbon sources [29]. Nevertheless, at pH 3, *L. plantarum* declined in growth over time, which indicates that the microbe is not at optimum conditions at that pH level; [30] were able to establish the optimum pH for *L. plantarum* to be close to 6 at 30 °C. This study looked at the ability of the microbe to tolerate various pH levels at 37 °C. Thus, at that temperature, *L. plantarum* seems to do very well at a range of pH 4 to pH 7.

*S. boulardii* also showed exceptional tolerance to acidic conditions with a significant difference (*p* < 0.05) in growth only at pH 2, where a decrease in CFU is seen from 5.18 log CFU/mL at 0 h to 4.56 log CFU/mL at 5 h (Figure 1). However, at pH 3 to pH 7, the CFU remained steady at 5.1–5.2 log CFU/mL from time 0 h to 5 hrs. An earlier report of Graff et al. (2008) [31] demonstrated that in phosphate buffer of pH 6.8, *S. boulardii* viability remained stable for 6 h. They further reported that in HCl of pH 1.1, the viability of *S. boulardii* significantly decreased after incubation for 5 min or higher. When observed under scanning/transmission electron microscopy, morphological damages and rupture of the *S. boulardii* wall were observed at low pH, and the threshold value from which *S. boulardii* viability was unaltered was at pH 4. However, this project showed that *S. boulardii* did not increase in number after 4 h at pH 4 but rather remained constant.

Overall, *L. plantarum* proved to thrive better than *B. longum* and *S. boulardii* in acidic conditions as there were significantly more CFU/mL over time of *L. plantarum* at each pH level. However, at pH 3 and pH 2, both *L. plantarum* and *B. longum* showed a drastic decrease in acidic tolerance (*p* < 0.05), while *S. boulardii* remained almost constant at all pH levels over time. As indicated in vitro, these microbes should be able to withstand the acidic conditions of the various segments of the chicken GIT, starting from the mouth where the pH of the saliva ranges from 6.7 to 6.9 to the crop with a pH range from 4.4 to 4.9. Digestion in the chicken begins in the proventriculus, the true or glandular stomach, where there is a high concentration of acids for food breakdown, which results in the pH range of 4.0 to 4.4. *L. plantarum* and *S. boulardii* would do better in this area than *B. longum* based on the results of this project. Out of the three microbes, *S. boulardii* would be most effective in the gizzard, which has a pH of 2.6. Most importantly, these microbes should be able to survive in the GIT and, most importantly, in the small intestine, the area where most probiotic activity would be beneficial to the host. According to Sanhueza et al. (2015) [32], bacterial survival and acclimation in the GIT involve cellular changes permitting the survival of microorganisms to prolonged acid pH exposure.

Bile tolerance was evaluated by subjecting the potential probiotics’ microbes *B. longum*, *L. plantarum*, and *S. boulardii* to varying concentrations of bile beginning at 0 (control), 1, 1.5, 2, and 3% for 6 h. *B. longum* survived all bile concentrations, but the number of colonies recovered was significantly lower (*p* < 0.05) than that of the control (Table 2). For instance, at a bile concentration of 1% the *B. longum* colony recovery was 1.3 log CFU/mL, whereas the recovery at 3% bile was 1.64 log CFU/mL. Interestingly, as time progressed, *B. longum* increased in numbers in the higher bile concentrations of 3% and 2% such that at time 0–6 h, the *B. longum* count increased from 1.64–2.34 log CFU/mL (*p* < 0.05) and 1.58–2.09 log CFU/mL (*p* < 0.05), respectively. Nonetheless, this trend was consistent with the lower bile concentrations. It appears that *B. longum* was affected by the presence of bile and exposure time but not the various bile concentrations.

A previous report by An et al. (2015) [33] of *B. Longum* subjected to bile stress, demonstrated a 3.05-fold and 3.44-fold up-regulation of the transcription and translation of a gene encoding bile salt hydrolase (BSH, BN_536), which catalyzes de-conjugation of glycine- or taurine-linked bile acids, respectively. The de-conjugation of bile salts may play an important role in bile tolerance because of the detoxification properties. Therefore, BSH activity may be a desirable feature of a probiotic as it can increase its survival ability in gut conditions [34].

On the other hand, *L. plantarum* was also able to survive the various bile concentrations, and there was no significant in the bacterial count between the control and the bile containing cultures. While at lower exposure time to bile concentrations such as 1–2 h there were no significant differences in *L. plantarum* counts (6.61–7.42 log CFU/mL), a significant increase (*p* < 0.05) in the bacterial count (*p* < 0.05) was observed at all bile concentrations with increments in the incubation time from 1–6 h ranging from as low as 6.95 log CFU/mL in 3 h exposure time to as high as 8.65 log CFU/mL at 1.5% bile medium after 6 h incubation period.

Tolerance of *S. boulardii* to the bile concentrations was observed but with a slight reduction in the bacterial count with the introduction of the bile salts. As such, the *S. boulardii* count of the control ranged from 3.82 at 1 h post-incubation to 4.83 log CFU/mL at 6 h post-incubation. Regardless of the bile concentrations, *S. boulardii* survived all four bile concentrations with no significant reduction in bacterial cell count. However as observed with *B. longum* and *L. plantarum*, a significant increase in colony count of the *S. boulardii* was observed with an increase in incubation time (*p* < 0.05), especially at 1 and 1.5% bile concentrations. Therefore, all three microbes survived the various bile concentrations and their count increased over incubation time with the exception of *S. boulardii*. Notably, *L. plantarum* displayed significantly more survivability in bile than *B. longum* and *S. boulardii.* Hence, these microbes will not be affected by the bile concentrations in the GIT of chickens.

The potential for *B. longum*, *L. plantarum*, and *S. boulardii* to attach onto the intestinal mucosa of broiler chickens is presented in Figure 2. *L. plantarum* (Figure 2b,c) showed significant evidence of attachment to the GIT mucosa (blue arrows) when compared to the control (Figure 2a) with a clear mucosal surface. A closer assessment (Figure 2c) showed how the cell membrane seemingly remodels and forms what appears to be an envelope as *L. plantarum* buries itself into the cell for attachment. While in the control (Figure 2a) the surface of the mucosal cells is relatively clean and the surface of the cell membrane smooth, it is quite visible that *B. longum* (Figure 2d,e) is attached to the luminal surface of the GIT. Figure 2e shows the irregularity in the cell surface around the area where the bacteria are attached. Upon closer examination, cytoskeleton rearrangement was observed as shown by the arrow. This demonstrates an interaction between *B. longum* and the cellular cytoskeleton to accommodate its attachment to the intestinal mucosa. It is often seen that when bacteria attach to the surface of an epithelium, the underlying cytoskeleton rearranges [35].

Unlike *L. plantarum* and *B. longum*, *S. boulardii* does not seem to have a strong attachment to the GIT mucosa. Figure 2F shows that even after washing the tissue, *S. boulardii* remained on the surface, suggesting that there is some form of attachment of the microbe to the mucosal cells. However, as there seemed to be no cell surface remodeling or exposure of cytoskeleton, it can be assumed that the attachment might not have been as strong as that of *L. plantarum* and *B. longum*. The intestinal mucosa is in continuous contact with luminal contents and is home to the microbiome, which commands a very high volume of activity in the GIT, mostly in the form of competitive exclusion. Therefore, it is expected that to establish a bacterial strain in the host’s GIT permanently, the microbe must be able to attach to intestinal mucosal cells [36]. Furthermore, it was reported earlier that many pathogens cannot apply their deleterious effects on the gut unless they are attached [37], and the beneficial action of probiotics has been attributed by their supposed ability to impede the adherence of pathogens to GIT mucosal cells [38].

The potential for *B. longum*, *L. plantarum*, and *S. boulardii* to competitively exclude pathogenic microorganisms in the GIT of broiler chickens was measured using the agar spot test (Table 3). The *L. plantarum* and *B. longum* both displayed positive inhibitory tests against *E. coli* O157:H7, *E. coli* 25922, *E. coli* 11775, *S. pullorum*, *S. enteritidis*, *S. typhimurum*, *E. faecium*, *E. faecalis*, *C. coli*, *C. lari,* and *C. jejuni*.

However, *S. boulardii* only showed positive inhibition against *C. lari* and *C. jejuni*. The *L. plantarum* zone of inhibition (ZOI) ranged from 30 to 34 mm against the *E. coli* strains (Table 3), 31–44 mm against *Salmonella* species, 24–33 mm against *Enterococcus* species, and 18–30 mm against *Campylobacter* species.

Conversely, *B. longum* showed positive ZOI against the pathogens with average ranges of 15–22 mm against the *E. coli* strains, 14–23 mm against *Salmonella* species, 16–20 mm against *Campylobacter* species, and 24 mm against *E. faecalis*. However, the results also showed insignificant inhibition properties of *B. longum* against *E. faecium* with a ZOI of 4.5 mm. In this study, *S. boulardii* significantly inhibited the growth of *C. lari* and *C. jejuni*, where a positive zone of inhibition of 12 mm was observed. However, *S. boulardii* showed no sign of the ability to inhibit the other selected pathogenic microbes.

Overall, although both *L. plantarum* and *B. longum* showed remarkable antagonism towards the pathogens, *L. plantarum* appeared to be more effective as its ZOI was significantly higher (*p* < 0.05) than *B. longum* as shown in Table 3. However, Santini et al. (2010) [24] reported that *B. longum* PCB 133 was more effective than *L. plantarum* PCS 20 against three strains of *C. jejuni*. They also found that *B. longum* PCB 133 possessed interesting probiotic properties and a marked anti-campylobacter activity both in vitro and in vivo. That makes *B. longum* PCB 133 an excellent poultry feed additive candidate for the reduction of foodborne campylobacteriosis in humans.

The findings in this project also correspond with those of Asha (2012) [39] who found that *Lactobacillus* sp., isolated from chicken intestine, demonstrated inhibitory activity of 12.5 mm to 18.0 mm against *S*. *enteritidis*, *S*. *pullorum*, *S*. *typhimurium*, *S*. *blockley,* and three serotypes of *E*. *coli*. A previous study on the influence of goat and cow milk fermented by *Bifidobacterium longum* Bb-46 on the pathogenic *Salmonella enteritidis* D strain showed the highest degree of inhibition on the in vitro growth of *S. enteritidis* [40].

Furthermore, although *S. boulardii* was seemingly not as effective as *L. plantarum* and *B. longum* in this in vitro study that does not necessarily mean that *S. boulardii* does not have other antagonistic characteristics towards pathogens or protective attributes towards the host against the effects of pathogenic microbes. In a detailed review, Im and Pathoulakis (2010) [41] pointed out that the main mechanisms of action of *S. boulardii* include the inhibition of activities of bacterial pathogenic products, trophic effects on the intestinal mucosa, as well as modification of host signaling pathways involved in inflammatory and non-inflammatory intestinal diseases.

## 4. Conclusions

Based on the conditions of this study, *B. longum*, *L. plantarum*, and *S. boulardii* exhibited characteristics in vitro that suggest that they could be effective probiotics for poultry. Their ability to survive the various pH and bile concentrations indicate that they will survive the gastrointestinal tract environment of chickens. The remarkable changes in the mucosal cell surface and cytoskeleton associated with *B. longum* and *L. plantarum* attachment in vitro is evidence that these microbes can colonize the GIT of chickens. Moreover, these mucosal changes in cell structure suggest interactions between the microbes and the intestinal mucosa. Most profoundly, *B. longum* and *L. plantarum* were able to positively inhibit the growth of pathogens that cause problems in the poultry industry. These findings reaffirm the probiotic potential for these microorganisms in poultry production.

## Figures and Tables

**Figure 1 microorganisms-10-00676-f001:**
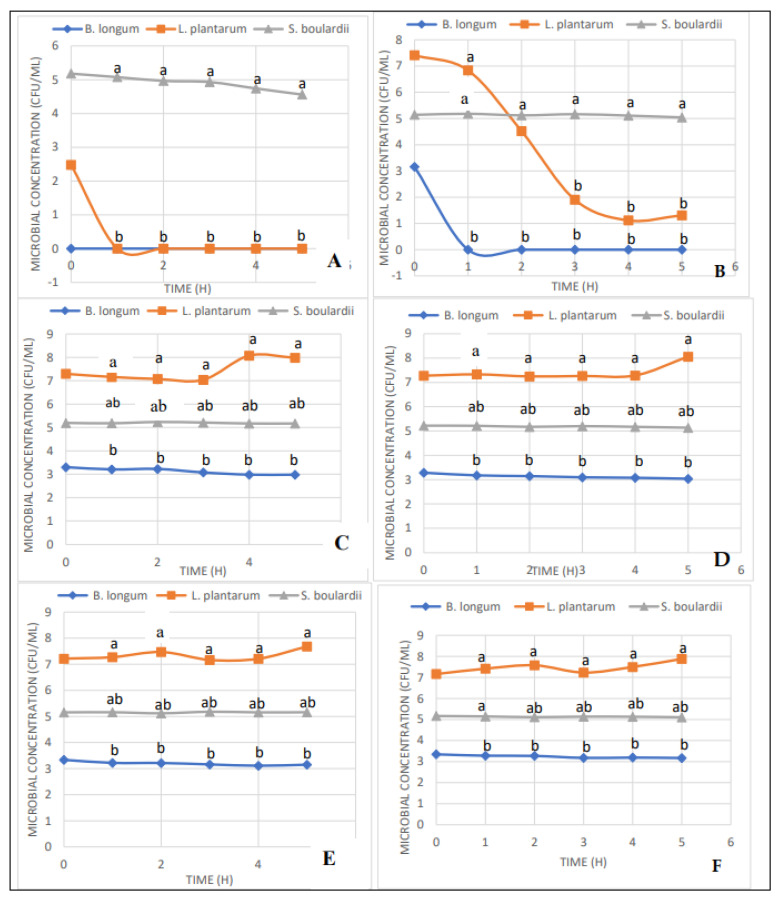
Concentration (CFU/mL) of *B. longum*, *L. plantarum* and *S. boulardii* subjected to pH conditions ranging from 2 (**A**), 3 (**B**), 4 (**C**), 5 (**D**), 6 (**E**) and 7 (**F**) to evaluate acid tolerance of these microorganisms. ^a,b^ Mean comparisons among the microorganisms with no common superscript differ significantly (*p* < 0.05).

**Figure 2 microorganisms-10-00676-f002:**
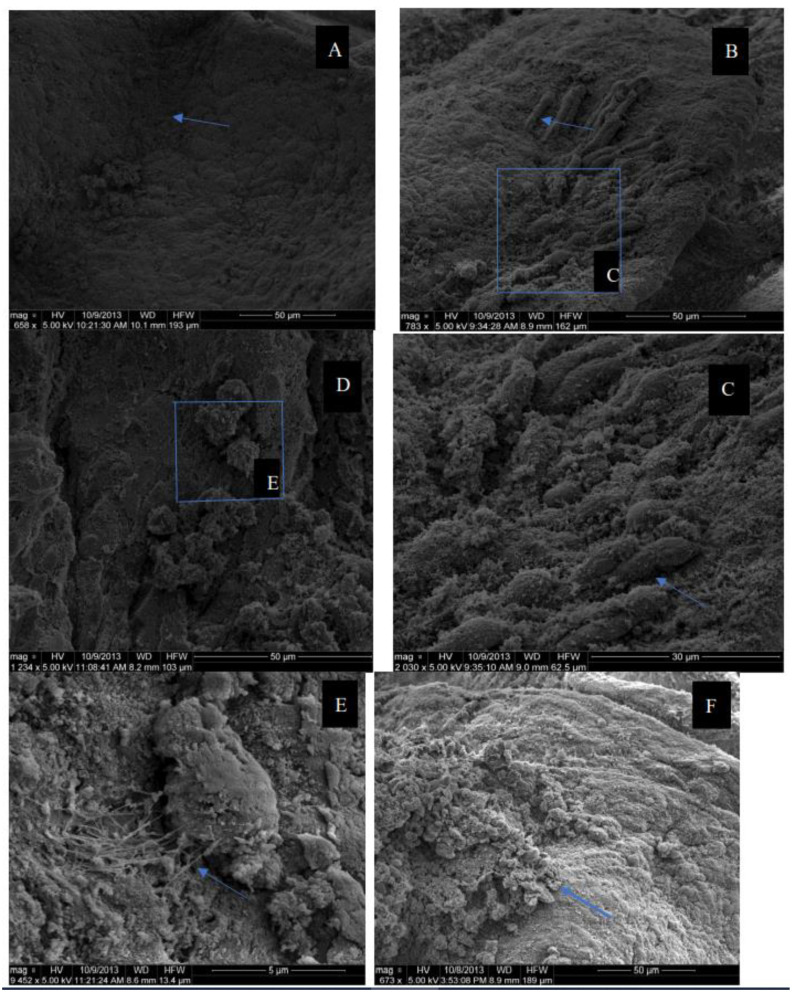
Scanning electron micrograph of probiotics attached to chicken intestinal mucosa. The control (**A**) is devoid of probiotics whereas probiotics treatments of birds fed the probiotics *Lactobacillus plantarum* (**B**,**C**), *Bifidobacterium longum* (**D**,**E**) and *Saccharomyces boulardii* (**F**).

**Table 1 microorganisms-10-00676-t001:** Pathogenic microbes and their culture conditions.

Genus	Species	Growth Conditions	Media
*Salmonella*	*enteriditis*, *pullorum*, *typhimurium*	Aerobic, 18–24 h at 37 °C	Tryptic Soy Broth (TSB)/Tryptic Soy Agar (TSA)
*Campylobacter*	*jejuni*, *coli*, *lari*	Microaerophilic, 48–72 h at 42 °C	Bolton Selective Enrichment Broth w/supplement and 5% horse blood customized agar *
*Enterococcus*	*Faecalis, faecium*,	Aerobic, 20–24 h at 35 °C	Tryptic Soy Broth (TSB)/Tryptic Soy Agar (TSA)
*Escherichia coli*	*E. coli* 0157:H7, *E. coli* 01: K1:H7, ATCC 25922	Aerobic, 18–24 h at 37 °C	Tryptic Soy Broth (TSB)/Tryptic Soy Agar (TSA)

Kizerwetter-Swida and Binek (2005) [25]. * Twenty percent bacteriological agar was added to Bolton broth as a solidifier.

**Table 2 microorganisms-10-00676-t002:** In vitro evaluation of survival of probiotics’ microorganisms in various concentrations of bile salts in a given period.

Bile Concentration (%)		3	2	1.5	1	0
Probiotic	Time (h)	Concentration (CFU/mL)				
*B. Longum*	1	1.64 ^bz^	1.585 ^byz^	1.425 ^bxy^	1.3 ^bx^	4.89 ^az^
	2	1.605 ^bz^	1.46 ^byz^	1.27 ^by^	1.19 ^bx^	4.92 ^az^
	3	1.625 ^bz^	1.41 ^bz^	1.285 ^by^	1.065 ^bx^	5.265 ^ay^
	4	1.81 ^byz^	1.52 ^byz^	1.17 ^by^	1.015 ^bx^	5.66 ^axy^
	5	2.065 ^bxy^	1.71 ^by^	1.355 ^bxy^	1.085 ^bx^	6.16 ^awx^
	6	2.345 ^bx^	2.09 ^bx^	1.82 ^bx^	1.43 ^bx^	6.675 ^aw^
*L. Plantarum*	1	7.285 ^az^	7.315 ^az^	7.385 ^az^	6.61 ^bz^	7.1 ^abz^
	2	7.395 ^ayz^	7.08 ^az^	6.96 ^az^	6.82 ^bz^	7.455 ^az^
	3	7.72 ^abcxy^	7.445 ^bcyz^	6.955 ^bcz^	7.245 ^cy^	8.165 ^ay^
	4	7.9 ^bx^	7.785 ^bxy^	7.35 ^bz^	7.7 ^bx^	8.705 ^ay^
	5	8.295 ^bw^	8.205 ^bcx^	8.025 ^cy^	7.975 ^cwx^	9.92 ^ax^
	6	8.35 b ^cw^	8.38 ^bcx^	8.655 ^bx^	8.14 ^cw^	10.15 ^ax^
*S. boulardii*	1	3.765 ^ax^	3.72 ^axy^	3.57 ^axyz^	3.61 ^ay^	3.82 ^az^
	2	3.815 ^abx^	3.59 ^by^	3.82 ^abwz^	3.72 ^abxy^	3.93 ^az^
	3	3.775 ^bcx^	3.785 ^bxy^	3.51 ^cy^	3.955 ^bxy^	4.33 ^ay^
	4	3.74 ^bx^	3.69 ^bxy^	3.81 ^bwx^	3.885 ^bxy^	4.545 ^axy^
	5	3.7 ^cx^	3.95 ^bx^	3.885 ^bcw^	4.065 ^bx^	4.72 ^awx^
	6	3.77 ^bx^	3.965 ^bx^	4.055 ^bw^	4.09 ^bx^	4.83 ^aw^

^abc^ Within probiotic means within rows with no common superscript differ significantly (*p* < 0.05). ^xyzw^ Within probiotic means within the column with no common superscript differ significantly (<0.05).

**Table 3 microorganisms-10-00676-t003:** Antimicrobial activity against enteropathogenic bacteria.

		Probiotics	
Enteropathogens	*L. plantarum*	*B. longum*	*S. boulardii*
	Zone of Inhibition (mm) *		
*E. coli* O157:H7	34.0 ^ay^	21.9 ^bx^	0 ^cy^
*E. coli* 25922	29.5 ^ay^	14.8 ^by^	0 ^cy^
*E. coli* 11775	30.0 ^ay^	14.6 ^by^	0 ^cy^
*S. pullorum*	43.6 ^ax^	22.8 ^bx^	0 ^cy^
*S. enteritidis*	31.3 ^ay^	13.6 ^by^	0 ^cy^
*S. typhimurum*	31.8 ^ay^	16.0 ^by^	0 ^cy^
*E. faecium*	24.0 ^az^	4.5 ^bz^	0 ^by^
*E. faecalis*	33.0 ^ay^	24.1 ^bx^	0 ^cy^
*C. coli*	17.8 ^az^	16.3 ^by^	0 ^cy^
*C. lari*	30.0 ^ay^	20.0 ^bx^	12.0 ^cx^
*C. jejuni*	22.0 ^az^	15.8 ^by^	12.0 ^cx^

* Zone of inhibition > 5 mm is positive inhibition. ^abc^ Means within a row with no common superscript differ significantly (*p* < 0.05). ^xyz^ Means within columns with no common superscript differ significantly (*p* < 0.05).

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
