# Peer review of "Evaluation of Selected Bacteria and Yeast for Probiotic Potential in Poultry Production"

_microorganisms, 2022, doi:10.3390/microorganisms10040676_

Round 1

Reviewer 1 Report

General Comments: In this report, Beverly Dixon et al. investigated the ability of the microbes to tolerate and survive varying pH levels and bile concentrations for the evaluation of the potential probiotic properties of L. plantarum, B. longum, and S. boulardii. The authors concluded that B. longum, L. Plantarum, and S. boulardii exhibited significantly higher CFU (P<0.05) at a pH range of 5 to 7, 4 to 7, and 2 to 7, respectively when compared with other pH levels. It was shown that L. plantarum had much higher colony-forming units per ml within each pH level, except at pH 2 where S. boulardii was the only microbe to survive over time. Moreover, it was demonstrated that L. plantarum and S. boulardii were able to tolerate  the various bile concentrations, B. longum and L. plantarum showed remarkable ability to attach to the intestinal mucosa and to inhibit pathogenic microbes.

There are some items in the manuscript that should be clarified. The following comments and questions are included in order to help the authors to improve the quality of the paper.

Minor Comments:

  1. Please modify the format Tables 1 and 2 as Tables 3.
  2. Please move the figure legends below the Figs.
  3. Please rearrange Fig 2 to make a clear connection of B and C (also, D and E).
  4. Lines 400-402: There are two papers included in reference 2.

The manuscript contains other grammatical and type errors. Please review the text carefully.

Author Response

  1. Please modify the format Tables 1 and 2 as Tables 3.

Response: Table 1 and 2 were modified as Table 3

  1. Please move the figure legends below the Figs.

Response: We followed the format of the journal unless that has changed, we may need some guidance from the editorial team.

  1. Please rearrange Fig 2 to make a clear connection of B and C (also, D and E).

Response: these figures were arranged this way because they are in a sequence and expounded from lower to higher magnification.

  1. Lines 400-402: There are two papers included in reference 2.

Response: The additional paper was deleted.

Reviewer 2 Report

see the attached file for reviewer remarks. space was allowed here for a portion of the remarks to be inserted

also, since only one attached file was allowed, an edited pdf document was sent to the editor

Author Response

  1. Comment: Suggested title change: Evaluation of Selected Bacteria and Yeast for Probiotic Potential in Poultry Production

Response: Title was changed as suggested by reviewer to “Evaluation of Selected Bacteria and Yeast for Probiotic Potential in Poultry Production)

  1. Comment: Line 103: The assumption is that this initial screening is to be followed by application of a defined probiotic cocktail in poultry experiments. Include a statement at the end of the introduction addressing this potential application.

Response: The following sentence was added as suggested by the reviewer. “It is essential to identify conditions that will permit these microorganisms to thrive and enhance protection against establishment of bacterial pathogens in poultry GIT.

  1. Comment: Line 110: Were these commercial strains initially isolated from animal sources? If yes, name the sources. Often it is more advantageous to use probiotic isolates derived from the target species.

Response: Commercial strains used in this study were purchased form fisher scientific

 Comment: Line 110: In the M&M, give the company name and location for all instruments, materials, chemicals, and media used is this study.

Response: Company name and location for all instruments, materials, chemicals, and media used is this study have been added in the current version

 Comment: Line 113: Give the name of the anaerobic chamber, company information, and the gas composition

 Response: Anaerobic chamber, company information, and the gas composition have been added in the methodology

 Comment: Line 114: Give the Bifidobacterium longum culture time and temperature

           Response:  Bifidobacterium longum culture time and temperature were added as 18 hrs at 37o C, respectively

  1. Comment: Line 116 Give the quantity amount of "trace of FeCl3"; unless that was the only information given by the provider

 Response: The information stated was the was the only information given by the provider

  1. Comment: Line 127 Rewrite for clarity: 100 µL of culture per 2 mL of sterile PBS suspension. Was 100 µL of culture added to 2 mL of PBS?

Response: For each microbe, 100µl of culture per 2mL of sterile PBS suspension was added separate-ly into a series of 6 tubes.  Was rewritten as “For each microbe, 100µl of culture was added separately to 2mL of sterile PBS suspension in a series of 6 tubes”

  1. Comment: Line 128. Give the medium volume and the inoculum CFU, or the final CFU/mL, for the six tubes.

Response: The final CFU/mL varied at different pH values as displayed in Figure 1

  1. Comment: Line 129. The pH range 0.5, 1, 2, 3, 4, and 5 differs from those on Line 204, pH 2, 3, 4, 5, 6, and 7

        Response:  The pH range” 0.5, 1, 2, 3, 4, and 5”, others pH 6, 7 values originally left   

        out in the   first draft are now added.  Only pH 2, 3, 4, 5, 6, and 7 were selected and   

        displayed in the study.

  1. Comment: Line 141 Give the Lactobacillus plantarum, Bifidobacterium longum, and Saccharomyces boulardii time 0 inocula in CFU/mL and the final CFU/mL in the inoculated media.

 Response:  At time 0 for all microbes, final CFU/mL varied at different pH values as displayed in Figure 1

  1. Comment Line: 143 Were separate assays run with bile and with Sigma ox gall, or was the bile derived from ox gall?

 Response: Only sigma ox gall was used in this study

  1. Comment: Line 155: Give the broiler breed and sex?

Response: Male Ross 308 broilers

  1. Comment Line 155: Were chick boarding, handling, and treatment protocols reviewed and approved by the appropriate institutional animal care and use committee?

       Response: All norms or ethics for protection and animal welfare were approved by the          

       Institutional Animal Care and Use Committee (IACUC) at Tennessee State University 

      (TSU).

 Comment Line 163 Were the duodenal segments inoculated with the strains externally or internally in the lumen?

      Response: duodenal segments were inoculated with the strains externally

 Comment line 167: Give the rpm or the method of moderate agitation

 Response: 270 rpm

  1. Comment line 175 Give the city and state for VUMC

Response: Nashville, TN has been added to the manuscript.

  1. Comment line 183: What is meant by "stubs"? Define or describe and give manufacturer information.

 Response: A stub is metal item upon which the sample is mounted for imagining.

  1. Comment line 184:  Sputter-coated with what and how? 191 Give the OD reading and the CFU/mL for A600 of about 0.7 mL

 Response: The samples were sputter-coated with gold using a sputter coater.

 Comment line 191: Give the OD reading and the CFU/mL for A600 of about 0.7 mL

Response: We did not get the OD reading and CFU/mL for A600 of about 0.7 mL at the time of the experiment

 Comment line 194: The current naming convention is to place Salmonella in italics and the serotype in regular font and with the first letter in capital

Response: The naming conversion was corrected as suggested by the reviewer: as S.   Pullorun, S. Typhimurum, S. Entritidis

22. Comments line 200: Were the dots replicated on each plate?

Response: yes, the dots replicated 3 times on each plate

  1. Comment line 200: A Statistical Analysis paragraph is needed.

Response: Statistical analysis paragraph was added; line 214-217 of the revised manuscript.

  1. Comment line 205: Do not use the word significant unless it is related to statistical comparison and accompanied by a P-value.

Response: The word “significant’ was substituted with “higher”, Line 222 of the revised manuscript.

  1. Comment line: 207 Is 2.98 CFU/mL a log-transformed number?

Response: 2.98 CFU/mL is a log-transformed number; and “log” was added to all transformed numbers CFU/mL

  1. Comment line 21:1 Is "slight decline" a statistical (P < 0.05) or relative (P > 0.05) reference?

Response: the sentence was changed to “….a slight but not statistically significant decline….” Line 228 of the revised manuscript.

  1. Figure 1 Use notation in the graphs for each strain to indicate statistically significant differences.

Response: Notations are now used to indicate statistical differences among the strains concentration at various pH levels.

  1. Comment line:219: Is "significant survivability" a statistical reference supported by P < 0.05?

Response: the statement was changed to “The Lactobacillus plantarum, on the other hand, displayed tremendously significant survivability (P<0.05) at pH 4-7 when compared to other microorganisms.” Yes, the differences were statistically significant at P<0.05 as shown in Figure 1.

  1. Comment line 231 If petrol is used as term for gasoline, replace it with the USA term gasoline.

Response: “petrol” was changed to “gasoline”

  1. Comment line 237: Is "significant difference" a statistical reference supported by P < 0.05?

Response: Yes, it is and P<0.05 was added to the statement.

  1. Comment line 247: Is "remain constant" a statistical reference supported by P > 0.05?

Response: This was not a comparison among the microorganisms, but an observation of the behavior of the individual microorganisms in response to pH and time of exposure to the pH.

  1. Comment line: 250 Is "drastic decrease" a statistical reference supported by P < 0.05?

Response: The decrease in acid tolerance was significant (P<0.05) and this was noted in the revision.

  1. Comment line 270 Is "increased in numbers" a statistical reference supported by P < 0.05?

Response: the increase in numbers due to time of exposure to bile was significant and therefore “P<0.05” was added in the text to denote statistical differences.

  1. Comment line271 Is "count increased" a statistical reference supported by P < 0.05?

Response: Yes, but not with S. boulardii. The statement was changes to: Therefore, all three microbes survived the various bile concentrations and their count increased over incubation time with the exception of S. boulardii.

  1. Figure 1 Use the same y-axis and x-axis scale for all the graphs. Visual

comparisons of information offered in graphs with mis-matched axis scales can be misleading.

Response: The scaling of the graphs was retained because it was automatically configured in developing the graphs. The letters denoting the statistical differences between the counts of the microorganisms can now address any concerns on the graphs.

  1. Comment: Table 2 Are the numbers log-transformed?

Response: In Table 2, all numbers are log-transformed

  1. Comment: Table 2 Were all CFU/mL at time 0, for each respective microbe, the same for all bile concentrations?

Response: Yes, they were the same

  1. Comment: Table 2 Round out decimals to two paces.

Response: Decimals were rounded to two places

  1. Comment: Table 2 Reverse the concentration row numbers to ascending order.

Response: The concentrations were reversed to ascending order

  1. Comment: Table 2 Place the statistical letters in superscript font and not in smaller sized font.

Response: Statistical letters were placed in superscript font

  1. Comment: Table 2 There is no stand-alone x superscript for time 0.

Response: “w” is the standalone superscript in the comparison.

  1. Comment: 286 6.61 and 6.82 at 1% were significantly different from values at 1.5% to 3%.

Response: Yes, this was noted and P<0.05 added to denote the statistical differences.

  1. Comment line 294: See the attached Bile Differences table in relation to:

"However as observed with B. longum and L. plantarum, an increase in colony counts of

the S. boulardii was observed with an increase in incubation time, especially at 1 and

1.5% bile concentrations."

Response: the statement is accurate and “a significant” and “P<0.05” were added to the statement.

  1. Comment line 30:2 Use a more distinctive lettering system to label the photographs and the inserts.

There are too many images labelled C and E to track without confusion.

Response: the images were laid out in this manner to make it easier for readers to follow in sequence.

Figure 2 Lighten photograph 2A and 2D for better visibility.

Response: The image was replaced

References Use the following reference format:

Author 1, A.B.; Author 2, C.D. Title of the article. Abbreviated Journal

The references were revised accordingly.